# Delineating the SARS-CoV-2 Induced Interplay between the Host Immune System and the DNA Damage Response Network

**DOI:** 10.3390/vaccines10101764

**Published:** 2022-10-20

**Authors:** Christina Papanikolaou, Vasiliki Rapti, Dimitris Stellas, Dimitra T. Stefanou, Konstantinos Syrigos, George N. Pavlakis, Vassilis L. Souliotis

**Affiliations:** 1Institute of Chemical Biology, National Hellenic Research Foundation, 11635 Athens, Greece; 2Third Department of Medicine, Thoracic Diseases General Hospital Sotiria, School of Medicine, National and Kapodistrian University of Athens, 11527 Athens, Greece; 3First Department of Medicine, Laiko General Hospital, School of Medicine, National and Kapodistrian University of Athens, 11527 Athens, Greece; 4Vaccine Branch, Center for Cancer Research, National Cancer Institute, Frederick, MD 21702, USA

**Keywords:** COVID-19 pandemic, SARS-CoV-2 coronavirus, aberrant immune response, DNA damage response, impaired interferon signaling, hyper-inflammation, delayed adaptive immune responses

## Abstract

COVID-19 is an infectious disease caused by the SARS-CoV-2 coronavirus and characterized by an extremely variable disease course, ranging from asymptomatic cases to severe illness. Although all individuals may be infected by SARS-CoV-2, some people, including those of older age and/or with certain health conditions, including cardiovascular disease, diabetes, cancer, and chronic respiratory disease, are at higher risk of getting seriously ill. For cancer patients, there are both direct consequences of the COVID-19 pandemic, including that they are more likely to be infected by SARS-CoV-2 and more prone to develop severe complications, as well as indirect effects, such as delayed cancer diagnosis or treatment and deferred tests. Accumulating data suggest that aberrant SARS-CoV-2 immune response can be attributed to impaired interferon signaling, hyper-inflammation, and delayed adaptive immune responses. Interestingly, the SARS-CoV-2-induced immunological abnormalities, DNA damage induction, generation of micronuclei, and the virus-induced telomere shortening can abnormally activate the DNA damage response (DDR) network that plays a critical role in genome diversity and stability. We present a review of the current literature regarding the molecular mechanisms that are implicated in the abnormal interplay of the immune system and the DDR network, possibly contributing to some of the COVID-19 complications.

## 1. Introduction

Coronavirus Disease 2019 (COVID-19) is an infectious disease caused by the Severe Acute Respiratory Syndrome Coronavirus 2 (SARS-CoV-2), a novel coronavirus that emerged in the city of Wuhan, China at the end of 2019. Being highly transmissible, the disease was rapidly spread worldwide and a few months later, in March 2020, the World Health Organization (WHO) declared the COVID-19 outbreak a global pandemic [1]. Unlike other coronaviruses that led to large-scale outbreaks (e.g., the SARS-CoV epidemic occurred in 2002, later eradicated; the MERS epidemic: firstly, reported in 2012, still ongoing), COVID-19 has shaped the human history of the 21st century and continues to pose unprecedented challenges for healthcare systems and socioeconomic structures globally [2]. COVID-19 is characterized by an unpredictable and extremely variable disease course ranging from asymptomatic cases to severe illness that can lead even to death. While upper respiratory symptoms are the most common acute manifestations encountered in the majority of patients, many of them develop interstitial pneumonia that may progress to respiratory failure and acute respiratory distress syndrome (ARDS) requiring mechanical ventilation and admission to the intensive care unit (ICU) [3]. Although SARS-CoV-2 predominantly causes pulmonary disease, a wide spectrum of extra-pulmonary clinical manifestations has been also observed. Literature suggests that any system (hematologic, cardiovascular, renal, gastrointestinal and hepatobiliary, endocrinologic, neurologic, ophthalmologic, dermatologic system) can be affected and the main components of SARS-CoV-2 ability to provoke multiple organ injury are (i) direct virus-induced cytotoxicity in angiotensin-converting enzyme 2 (ACE2) expressing cells, (ii) dysregulation of the renin-angiotensin-aldosterone system resulting from virus-mediated ACE2 downregulation related to virus entry, (iii) immune dysregulation, (iv) endothelial cell injury and thrombo-inflammation, and (v) tissue fibrosis [4].

Previous studies have shown that coronavirus infection can cause DNA damage in host cells and thus activate the DNA damage response (DDR) network, a well-organized network of molecular pathways that plays an important role in genome stability and diversity [5]. On the other hand, the failure to respond to the virus-induced DNA damage can result in a variety of human diseases, since DNA damage may lead to mutagenesis and genomic instability. Herein, we present a review of the current literature regarding the molecular mechanisms employed by SARS-CoV-2 towards (a) the abnormal activation of the immune system and (b) the induction of DNA damage and aberrant mechanisms of DNA repair, possibly contributing to some of the COVID-19 consequences.

### 1.1. Immune Dysregulation

Immune dysregulation, the hallmark of COVID-19 disease course and severity, has been a major scientific focus and the identification of the precise immunopathological mechanisms remains elusive. Evidence supports that aberrant immune response to SARS-CoV-2 has been mainly attributed to impaired interferon (IFN) signaling, hyper-inflammation, and delayed adaptive immune responses [6]. In brief, impaired type I and type III IFN induction offers fertile ground for rapid virus replication and subsequent pathogen recognition receptor (PRR)-induced abnormal inflammatory response. Hence, multiple immune cells, such as endothelial cells, macrophages, monocytes, dendritic cells, and T-cells, are activated, stimulating various inflammatory pathways, leading to the excessive production of cytokines and chemokines (e.g., IL-1β, IL-8, IL-6, tumor necrosis factor). Although hyper-inflammation inhibits further virus spread, it progressively results in tissue damage and multi-organ failure. Simultaneously, after SARS-CoV-2 entry into cells, coronavirus-encoded viral proteins activate NLRP3 (NOD-, LRR- and pyrin domain-containing protein 3) inflammasomes, which stimulate NF-kB (nuclear factor-kappa B) signaling, caspase-1 activation, interleukin-1β (IL-1β), and interleukin-18 (IL-18) cleavage into their active forms and pyroptosis initiation, a highly inflammatory form of lytic programmed cell death. Furthermore, the complement system, a critical effector for pathogen recognition and elimination, contributes to the endotheliitis and thrombosis observed in COVID-19 and enhances disease. It is supposed that anaphylatoxins C3a and C5a promote neutrophils activation and the interaction between the complement and the neutrophil extracellular traps reinforces complement cascade. The adaptive immune system commences soon after the innate immune responses and the antibody-producing B cells, together with CD4^+^ T cells and CD8^+^ T cells aid in infection control by promoting virus clearance and providing protection against it through cytokines secretion. Nonetheless, CD4^+^ and CD8^+^ T cells are significantly reduced in patients with severe SARS-CoV-2 pneumonia [3,6,7].

### 1.2. Impaired Interferon Induction

Interferons are essential cytokine mediators that regulate host antiviral response; they inhibit virus entry, replication, translation, and egress and they stimulate immune cells recruitment and proliferation [6]. A finely tuned antiviral response, represented by type I/III IFN-mediated responses that precede pro-inflammatory ones, is crucial for maintaining a balance for optimal protection and minimal host damage [8,9]. Any deviation from this balance can induce paradoxical hyper-inflammation with disastrous consequences for human health.

In the context of COVID-19, IFN signaling has been under intensive investigation and debate. Several studies suggest the duality in the role of the IFN pathway, since both protective and deleterious effects have been documented [10]. Specifically, its beneficial role is presumed by the ample evidence that severe COVID-19 is characterized by diminished or suppressed IFN production and activity, inborn errors associated with the IFN signaling cascade or the presence of autoantibodies against IFNα or IFNγ [10,11,12,13,14,15,16,17,18,19]. For instance, in contrast to mild or moderately ill patients, 10% of cases with life-threatening COVID-19 pneumonia were found to have anti-IFN-1 autoantibodies and almost 4 in 10 patients had genetic defects of Toll-like receptor 3 (TLR3)- and interferon regulatory factor 7 (IRF7)-dependent type I interferon immunity [14,16]. Notably, auto-Abs neutralizing type I IFNs were even present in a proportion of vaccinated patients that developed a breakthrough hypoxemic COVID-19 pneumonia, despite two mRNA vaccine inoculations and the presence of circulating antibodies capable of neutralizing SARS-CoV-2 [20]. Lastly, in the case of moderate to severe hospitalized patients, IFN-λ and type I IFN secretion were both decreased and delayed, induced only in a fraction of them, as they became critically ill [12]. The aforementioned eventually highlights the crucial role of IFN signaling in COVID-19 etiopathogenesis and progression.

Like other coronaviruses, SARS-CoV-2 employs various immune-escape strategies in order to limit competent IFN production and inhibit IFN signaling [21]. Four non-structural proteins (NSP) of SARS-CoV-2, including NSP13, NSP15, open reading frame (ORF) 7b, and ORF9b, have been reported to interact with host proteins involved in IFN signaling and impede IFN-1 mediated innate immune responses [22,23]. Moreover, at least eight proteins, namely NSP1, NSP3, NSP12, NSP13, NSP14, ORF3, ORF6, and M, have been additionally recognized as potent IFN-β inhibitors [24]. Among them, SARS-CoV-2 main protease, also known as M^pro^, 3CL^pro^ or NSP5, has been shown to suppress IFN production by preventing TRAF3-TBK1/IKK∈ complex formation, downregulate interferon-stimulated gene (ISG) induction and target retinoic acid-inducible gene I (RIG-I)/melanoma differentiation-associated protein 5 (MDA5) signaling, thus attenuating antiviral immunity, and enhancing viral replication [25,26]. Similarly, ORF6 and NSP3 restrain IRF3 nuclear translocation and the cleavage of ISG15 from IRF3, respectively, whereas NSP1 effectively obstructs RIG-I-mediated IFN responses by binding to 40S ribosomal subunits and blocking host mRNA translation [24,27,28]. NSP1 and NSP6 also interfere with STAT1/STAT2 phosphorylation and/or their nuclear translocation resulting in the suppression of IFN signals [29]. Likewise, the S protein inhibits STAT1 phosphorylation and its nuclear translocation by impeding JAK-STAT1 interaction [30]. Another mechanism of immune evasion is through MHC class I pathway modulation. NLRC5, an MHC class I transactivator, is suppressed both transcriptionally and functionally by ORF6 protein through (i) type II IFN-mediated STAT1 signaling, with subsequent NLRC5 and IRG1 gene expression blockage, and (ii) NLRC5 nuclear import inhibition. All the above lead to the conclusion that there is a complex regulatory network between SARS-CoV-2 and the innate immune system, in which host immunity downregulation and evasion are mainly driven by direct disruption of antiviral-associated proteins [31].

IFN kinetics during SARS-CoV-2 infection is not yet clear and an interpatient variability is presumed [3,32]. While a robust IFN response that coincides with viral replication peak, promotes its elimination, and declines as the virus is cleared characterizes mild disease course, IFN-I and –III responses are milder and delayed relative to viral proliferation as the disease progresses. The above allows persistent viral presence and prolonged IFN and inflammatory cytokines expression that triggers immune-mediated pathogenesis [32]. Hadjadj and colleagues [11] demonstrated that IFN-I signaling was highly dysregulated in severely or critically ill COVID-19 patients, as indicated by low IFN-I and ISGs levels, despite increased levels of TNF-, IL-6-, and NF-κB-driven inflammatory responses. In the same context, Lee and colleagues [33] reported that in severely ill patients, all the PBMC cell types displayed hyper-inflammatory signatures, marked by a TNF/IL-1β-mediated inflammatory response. Furthermore, in those patients, IFN-I response co-existed with the TNF/IL-1β-driven inflammation, whereas it was absent in milder cases. Hence, aberrant IFN kinetics is presumed to play a pivotal role in the excessive inflammation seen in severe COVID-19 patients.

### 1.3. Hyper-Inflammation

Cytokine storm is a life-threatening condition accompanied by excessive cytokines/chemokines production and immune cell hyper-activation that can be triggered by various causes, such as pathogens, autoimmune disorders, or malignancies. It involves an immune response that causes collateral damage and may overweigh the immediate benefit of the immune response [34].

Since the pandemic’s onset, it became clear that hyper-inflammation is a key component of COVID-19 etiopathogenesis and it is integrally linked to lung injury, multi-organ failure, and mortality [7,34,35]. In serum samples collected by patients with cytokine storm, increased levels of IL-1β, ΙL-6, tumor necrosis factor (TNF), macrophage inflammatory protein (MIP) 1α and 1β, IFN-γ, inducible protein 10 (IP-10), and vascular endothelial growth factor (VEGF) were detected [36,37].

The underlying mechanism of cytokine storm in COVID-19 is complex. SARS-CoV-2 is a cytopathic virus that causes cell death via pyroptosis [38]. Upon virus invasion, the components (ATP, DNA, etc.) released by the lysis of the infected cells can be detected as DAMPs (damage-associated molecular patterns) by the pattern recognition receptors (PRRs) and diverse immune cells (endothelial cells, macrophages, monocytes, dendritic cells, natural killer T cells), and various inflammatory pathways are activated. Normally, the above would favor virus clearance. However, in the case of moderate or severe COVID-19, high viral load and/or individual immunogenetic factors alter the immune landscape; low levels of antiviral interferons and several cytokines (IL-1β, IL-2R, IL-6, IL-7, IL-8, IL-17, and TNF-α) and chemokines (CCL-2, CCL-3, CCL-5, CCL-7, CXCL-10) are produced, and a systemic or/and diffuse pulmonary hyper-inflammatory state develops as a sequela [39,40]. IL-1, TNF-α, and IL-6, the three most important pro-inflammatory cytokines responsible for immune system activation, are thought to prominence in the cytokine storm, associated with “viral sepsis syndrome” observed in critically ill patients and be prognostic indicators of poor outcomes [41,42]. Moreover, the pathways that have been proven to contribute to cytokine storm progression so far are the impaired IL-6/Janus kinase/signal transducer and activator of transcription (IL-6/JAK/STAT) signaling pathway, the interferon cell signaling cascade, the TNF-α-nuclear factor-kappa B (TNFα/NF-κB) pathway, the toll-like receptor (TLR) pathway, the antibody-mediated pathway, Bruton tyrosine kinase (BTK) pathway, and the renin-angiotensin system (RAS) pathway [36,37].

Among the host immune cells that exhibit exacerbated activation and contribute to the establishment of a hyper-inflammatory state in COVID-19 patients, circulating monocytes and monocyte-derived macrophages predominate [43,44,45]. For instance, in blood samples of patients admitted to the ICU, a notable expansion of CD14^+^CD16^+^ monocytes featuring a high expression of IL-6 was detected [46]. Likewise, severe COVID-19 cases were characterized by decreased non-classical and intermediate monocyte subsets, as well as circulating classical monocytes expressing CD169, suggesting an increased activation status [44,47]. Furthermore, HLA-DR expression on monocytes, a reliable indicator of immunosuppression in critically ill patients, was reported diminished in hospitalized patients with severe COVID-19 pneumonia and was strongly correlated with disease severity, and the levels of soluble immunosuppressive factors, such as IL-10, TGF-β and VEGF [48,49,50,51]. So far, several hypotheses regarding the excessive activation of monocytes and monocyte-derived macrophages found in severe SARS-CoV-2 infection have been suggested. According to the prevailing one, delayed IFN responses and the subsequent insufficient viral clearance result in the continuous release of chemokines and GM-CSF from alveolar epithelial cells and the accumulation of immune cells (e.g., monocytes) into the lungs [50]. Thereafter, upon STAT pathway stimulation, monocytes differentiate into pro-inflammatory macrophages [45]. Simultaneously, monocyte-derived macrophages, potentially re-activated through TLR4 and TLR7 pathways, inducing oxidative stress reactions in the infected lung tissues [52,53]. Lastly, through the upregulation of SARS-CoV-2 entry receptors (e.g., ACE2, CD147) expression, IFN facilitates the viral access to the macrophage cytoplasm and the activation of NLRP3 inflammasome, thereby triggering IL-1β secretion [45]. The aforementioned cascade driven by monocytes and macrophages eventually reinforces the cytokine storm [50].

### 1.4. Delayed Adaptive Immune Response

Whilst aberrant innate immune responses are intrinsically involved in COVID-19 immunopathogenesis, the role of adaptive immunity has not been adequately explored. Adaptive immune responses play a crucial role in SARS-CoV-2 clearance during the later stages of the infection and seem to be major determinants of the clinical outcome, since they are relatively dysregulated in critically ill patients. Evidence suggests that early and robust interferon and adaptive immune responses are the main components of effective viral load and infection control, whereas impaired and prolonged interferon response alongside delayed and excessive cellular activation are associated with early inflammation and poor prognosis [54].

The protective role of the adaptive immunity in COVID-19 is well described. Detected as early as seven days after symptoms onset, cytotoxic CD8^+^ T cell response is correlated with effective virus elimination and milder course disease [55,56,57]. Profound lymphocytopenia, a common laboratory finding in patients with moderate or severe disease, is a predictor of the clinical outcome and its resolution contributes to recovery [58,59]. Furthermore, the beneficial role of cellular immunity is highlighted in several studies that compare T cell responses in asymptomatic, mildly symptomatic, or severely symptomatic individuals [6]. A highly functional SARS-CoV-2-specific cellular immune response accompanied by potent expression of T cell effector molecules IL-2 and IFNγ was identified in asymptomatic COVID-19 [60], whilst mild cases were characterized by early induction, the emergence of functional virus-specific T cells that target multiple epitopes and prolonged contraction of T cell responses [55,61,62]. Notably, in convalescent patients that presented mild disease course, virus-specific memory CD4^+^ and CD8^+^ T cells were present [55]. In the airways of the patients, the frequency of tissue-resident T cells that exhibit functionally protective phenotypes was linked to higher survival rates [63].

The excessive pattern of T-cell activation in COVID-19 has led to the hypothesis that T-cell-mediated immune responses may be part of the immunopathology. Although the presence of CD8^+^, which expresses high levels of effector molecules, improves the clinical outcome, the extreme, ongoing activation and the expression of markers of potential exhaustion (PD-1, Tim-3) may be detrimental [59,64,65]. Zhou and colleagues [46] have reported that after SARS-CoV-2 infection, a subset of CD4^+^ T cells is rapidly activated and differentiates into T helper 1, thus resulting in the secretion of high levels of granulocyte–monocyte colony-stimulating factor (GM-CSF), IL-6 and IFNγ. CD14^+^ CD16^+^ monocytes with abundant IL-6 expression are stimulated by the secreted cytokines and a hyperinflammatory state is established. Kusnadi and colleagues [66] revealed that severely ill patients display impaired exhaustion features in SARS-CoV-2-reactive CD8^+^ T cells compared to those mildly infected, since a higher frequency of CD8^+^ T-cells, which express predominately inflammatory cytokines/chemokines (CCL3, CCL4, TNF and others), and cytotoxic effector molecules (granzyme B, granzyme H, granulysin, etc.), and fewer T cell exhaustion associated molecules (TIM3, LAG3, CD38) were found.

SARS-CoV-2 infection induces neutralizing and non-neutralizing antibodies [67,68]. Neutralizing antibodies bind to the angiotensin-converting enzyme 2 (ACE2) receptor-binding domain (RBD) of the S protein and thereby limit the viral entry into the susceptible cells. The role of the non-neutralizing antibodies is not clearly understood, but it may involve antibody-dependent cellular cytotoxicity [69]. SARS-CoV-2 specific antibody response kinetics depends on disease severity and, in general, rapid antibody induction reflects reduced viral load and severity [67,70,71]. In convalescent patients, higher levels of virus-specific antibodies were identified in those who survived the severe disease, whilst a discordant T-cell and antibody response, defined as the induction of T-cell response with a lack of seroconversion, was noticed in asymptomatic and mild cases [72]. The above supports the assumption that a coordinated activation of the humoral and the cellular branch of adaptive immunity is necessary for effective SARS-CoV-2 infection control and determines disease severity. Additional data from macaque challenge experiments support contributions from both cellular and humoral immunity for optimal protection against disease [73].

## 2. The Impact of COVID-19 on Cancer Patients

In the general population, gender (men showed 1.57-fold higher odds ratio for mortality and a 1.65-fold higher for severe infection than women) [74], advanced age (median age > 60), obesity, and illnesses, such as congestive heart failure, coronary heart disease, diabetes, hypertension, hyperlipidemia, and cancer, are the main risk factors for severe COVID-19 [75]. The risk of COVID-19 is also influenced by race and ethnicity, with Black and Hispanic persons being more at risk than white people. Particularly, cancer patients were thought to be more vulnerable to SARS-CoV-2 infection than those without cancer, not only due to age, as cancer incidence is strongly correlated with advancing age, but also due to the high prevalence of cancer risk factors also associated with COVID-19, specifically abnormal thoracic computed tomography scans and smoking, as well as cancer-associated metabolic disorders like diabetes and hypertension. For cancer patients, there are both direct and indirect consequences of the COVID-19 pandemic [76]. Regarding the direct consequences, observations suggest that cancer patients are not only more likely to be infected by SARS-CoV-2, but are also prone to develop serious complications, such as admission to an intensive care unit (ICU), the need for mechanical ventilation, and even death [77,78,79]. Although all cancer types are associated with higher morbidity and mortality, lung cancer patients are vulnerable to dealing with COVID-19 complications as a sequela of the pre-existing risk factors or conditions, such as smoking-related lung damage, cardiovascular and respiratory comorbidities, and older age [80,81,82,83,84,85].

The susceptibility of cancer patients to longer and more severe COVID-19 courses is mainly attributed to their systemic immunosuppressive status, derived either by cancer itself or the anticancer treatment [86]. Indeed, cancer itself can change the way the immune system works [87]. An important cell type involved in immunosuppression in cancer is the CD4^+^FOXP3^+^ regulatory T cell (Treg), a T-cell subpopulation that suppresses abnormal immune responses to self- and nonself-antigens to maintain immune homeostasis [88]. In various types of malignancies, these cells accelerate immune evasion by tumor cells, thus resulting in the development and progression of cancer [89]. Moreover, cancer may lead to T-cell exhaustion, a state of T-cell dysfunction characterized by progressive loss of critical effector functions [90]. Importantly, tumor cells can metastasize in the bone marrow, preventing the synthesis of new immune cells. Bone marrow metastases are more common in patients with breast, lung, and prostate cancer [91]. Moreover, cancer cells evade attacks from the immune system by inhibiting antigen recognition and creating an immunosuppressive tumor microenvironment [87]. In addition, chemotherapy suppresses the bone marrow and causes lymphopenia, neutropenia, and thrombocytopenia, thus resulting in susceptibility to infections, such as SARS-CoV-2 [92,93,94]. Particularly, lymphopenia makes the patient highly vulnerable to viral infection, which becomes more pronounced when COVID-19-associated lymphopenia is added to the situation, producing severe disease. Radiation therapy also strongly favors lymphopenia because of direct exposure of lymphocytes to radiation, with increases risks in proton therapy, stereotactic body radiation (SBRT), or hypofractionated radiation therapy [95].

As for the indirect consequences, the spectrum of cancer care has been disrupted. Significant delays in cancer diagnosis and treatment have been reported since patients often delay or even postpone assessing cancer suspicious symptoms either for fear of SARS-CoV-2 exposure or even due to COVID-19. Moreover, access to health care services has been significantly limited as almost all healthcare workers are employed in the management of COVID-19 patients. Delays were also observed in cancer patients starting anticancer therapy or under follow-up [96,97,98]. Notably, patients who experienced either an interruption or failure to start therapy because of COVID-19 had a statistically shorter overall survival than those who remained on treatment [97].

During the COVID-19 pandemic, managing oncological surgeries is an additional significant challenge [99]. Particularly, for patients undergoing lung cancer surgery, the UK Lung Cancer Coalition’s Clinical Advisory Group reported that increased mortality rates were obtained in cancer patients who got infected with SARS-CoV-2 after lung cancer surgery [100,101]. An additional challenge in that context was the ICU unavailability since ICU beds had been dedicated to patients with COVID-19.

Data showing a more than 60% decrease in cancer clinical trials and biological therapies during the pandemic further highlights the impact of the COVID-19 pandemic on oncology research [102,103,104]. There are many reasons for the negative impact on cancer clinical trials, including travel restrictions, research staff being redeployed to frontline clinical activities and a significant decrease in the number of cancer patients visiting health units [105]. However, both patients’ visits to health centers and trial activations continuously recovered during the subsequent periods of the pandemic and, despite the ongoing nature of the pandemic, have now returned to almost normal levels [106].

## 3. DNA Damage Response and COVID-19

### 3.1. DNA Repair Mechanisms

Our cells develop DNA lesions on a daily basis. These lesions can inhibit basic cellular processes, such as genome replication and transcription, and if they are not repaired properly, they could result in mutations or genome aberrations, thereby posing a threat to the cell or even to the viability of a particular organism [107]. Several endogenous insults are responsible for forming these DNA lesions, including DNA base mismatch, oxidation, hydrolysis, and alkylation of DNA, as well as exogenous factors, such as ultraviolet and ionizing radiation and several chemical agents [108].

To protect against the genotoxic effects, cells have evolved several genome-protection pathways, collectively termed the DNA damage response network (Figure 1) [109]. DDR is an organized system that includes sensors, mediators, transducers, and effectors that activate various pathways, including DNA repair and cell cycle control. If the unrepaired DNA lesions are above a certain level, apoptosis or mutagenesis are triggered [110].

At least seven major DNA repair pathways are active throughout the cell cycle, with each one repairing different types of lesions.

(a)Nucleotide excision repair (NER). This mechanism repairs lesions that disrupt the DNA double-helix, such as bulky base adducts [111]. NER detects helix-distorting base lesions via two sub-pathways with different lesion detection mechanisms: transcription-coupled repair (TCR), which identifies lesions that inhibit transcription, and global-genome repair (GGR), which removes lesions throughout the genome.(b)Base excision repair (BER). This is a commonly used DNA repair process that identifies and repairs damaged DNA bases that do not alter the structure of the DNA helix. The cell uses BER to repair abnormal DNA bases, simple base-adducts, oxidative DNA damage and single-strand breaks (SSBs) [112]. There are two BER sub-pathways: the short-patch and the long-patch pathway. The activation of one or both of these two BER sub-pathways is determined by the origin of the damage and the cell cycle phase in which the damage occurs.(c)Mismatch repair (MMR). This pathway eliminates base substitution and insertion/deletion mismatches that occur when replication errors escape DNA polymerases’ proofreading function [113].(d)Homologous recombination repair (HRR). This is an error-free DSB repair mechanism that works throughout the S and G2 phases of the cell cycle to find a sister chromatid, which serves as a template to direct the repair of the damaged sequence [114].(e)Non-homologous end-joining (NHEJ). This mechanism repairs radiation- or chemically-induced double-strand breaks (DSBs), as well as intermediates of the V(D)J recombination and class-switch recombination (CSR) processes [115,116,117]. It is prone to errors and can function at any stage of the cell cycle. There are two subtypes of NHEJ: the canonical (c-NHEJ) and the alternative non-homologous end-joining (alt-NHEJ).(f)Interstrand cross-link (ICL) repair. This pathway repairs cross-links between the two strands of DNA, a critical event that usually results in cell cycle and replication arrest and eventually cell death [118]. In non-replicating cells, ICL repair is mediated by the NER mechanism, while in the S phase it is coupled to DNA replication and depends on the homologous recombination machinery [119].(g)Direct repair pathway. The only protein that is implicated in this mechanism is the O6-methylguanine-DNA methyltransferase (MGMT), which removes alkyl groups from the O6 position of guanine to a cysteine residue on itself and undergoes the degradation process [120].

### 3.2. COVID-19 and DDR

During the past few years, a wealth of information has been accumulated, shedding light on interactions between viral infections and the activation of DDR-related pathways [121]. For example, Chambers and colleagues [122] reported that the MMR pathway, a critical component of the DDR network, is required for the cellular anti-influenza A virus (IAV) response and controls the cellular fate following viral infection. Previous studies have shown that IAV infection leads to the death of infected cells through various cell death pathways, such as necrosis, necroptosis and pyroptosis, thus promoting effective virus clearance [123,124]. In addition to the IAV-induced death of infected cells, immune cells can effectively recognize and clear infected cells from the host, thus resulting in viral clearance from the host [125]. Interestingly, although IAV infection typically decreases cells’ MMR capacity, a subset of respiratory epithelial cells, named club cells, are remarkably capable of maintaining high levels of MMR activity [122]. Club cells’ increased MMR capacity efficiently removes the virus-induced oxidative DNA damage, thus allowing the transcriptional activation of antiviral genes, which probably aids in viral eradication and cell survival. In vivo, this has significant clinical implications because the loss of MMR activity reduced cell survival and exacerbated viral illness. In fact, Haque and colleagues [126] reported that a cancer patient with hereditary nonpolyposis colorectal cancer, a syndrome characterized by defective MMR, tested positive for SARS-CoV-2 for at least 54 days after the diagnosis of COVID-19. The authors proposed a connection between a deficient MMR mechanism and protracted viral shedding after SARS-CoV-2 infection, where the host repair system is harmed as a result of the virus-induced oxidative DNA damage and the impaired MMR.

Poly (ADP)-ribose polymerase (PARP) enzymes are a family of proteins that have been extensively investigated in many human diseases, including cancer, disorders of the central nervous system and RNA viral pathology [127]. Although the poly (ADP-ribosylating) (PARylating) PARPs catalyze the formation of branched or linear chains of ADP-ribose moieties and mainly function in the cellular response to DNA damage, several noncanonical mono(ADP-ribosylating) (MARylating) PARPs that modify their target proteins by the addition of a single ADP-ribose moiety, are implicated in cellular antiviral responses [128]. Interestingly, Heer and colleagues [129] have shown that SARS-CoV-2 infection induces MARylating PARPs, such as PARP7, PARP10, PARP12, and PARP14, and up-regulates the expression of genes that are encoding enzymes for salvage nicotinamide adenine dinucleotide (NAD) synthesis from nicotinamide and nicotinamide riboside, while down-regulating other NAD biosynthetic pathways. Importantly, PARP inhibitors had advantageous effects on SARS-CoV-2 infection by blocking the overactivation of macrophages and the cytokine storm that follows [39,40], as well as by preventing cell death [130]. Other studies have also shown that PARP inhibitors showed a protective role against the risk of COVID-19 progression in patients with cardiovascular, central nervous system, and metabolic diseases [131,132].

Furthermore, SARS-CoV-2 infection activated the DDR network in Vero E6, an African green monkey kidney cell line [133]. In that study, virus-infected Vero E6 cells exhibited (a) transcriptional upregulation of the Ataxia telangiectasia and Rad3-related (ATR) protein, (b) increased phosphorylation of Chk1 at serines 317 (S317) and 345 (S345), (c) increased phosphorylation of histone H2AX at serine 139 (S139; γH2AX), (d) decreased expression of the telomeric repeat-binding factor 2 (TRF2) subunit of the Shelterin system, a protein complex that plays a crucial role in telomere protection, while its absence results in DDR activation and the processing of chromosome ends by the DNA repair pathways [134], and (e) decreased telomere lengths. These findings suggest that SARS-CoV-2 affects telomere length, through the decreased expression of the TRF2 protein, thus triggering the DDR network mediated by the ATR signaling pathway. In addition, Sepe and colleagues [135] have shown that during aging the expression of the virus’ cell receptor ACE2 increased in mice and human lungs. They also reported that this increase was dependent on the DDR network, since both (a) the inhibition of the ATM kinase activity, resulting in the global DDR inhibition, and (b) the telomeric DDR inhibition through specific antisense oligonucleotides, prevented the upregulation of ACE2 following telomere shortening or the induction of DNA damage. Together, these results suggest that during aging, telomeric shortening or DNA damage activates the DDR network resulting in the upregulation of ACE2 and making older people more susceptible to SARS-CoV-2 infection.

In another study, the authors claim that SARS-CoV-2 infection of cells expressing high levels of ACE2, the SARS-CoV-2 spike protein induced the formation of syncytia and the generation of micronuclei due to DNA damage [136]. Interestingly, the authors reported that the formation of DNA damage within these syncytial micronuclei triggers the DDR network and the cGAS-STING-IFN signaling, through the recruitment of the γH2Ax and the cGAS proteins, thus resulting in cellular catastrophe and aberrant immune activation.

A recent study shows that exposure of *Poecilia reticulata* (a widely distributed tropical fish) adult fish to fragments of the SARS-CoV-2 spike protein induced genomic instability, DNA damage in circulating erythrocytes and induction of oxidative stress marked by increased levels of the antioxidant enzymes superoxide dismutase (SOD) and catalase (CAT) and accompanied by high levels of malondialdehyde (MDA), reactive oxygen species (ROS), and hydrogen peroxide (H_2_O_2_) [137].

Notably, previous studies have shown that NOX4-derived oxidative stress plays a crucial role in influenza virus proliferation [138]. Indeed, it has been demonstrated that lung epithelial cells infected with the influenza A virus experience a brief rise in intracellular ROS as a necessary phase in the development of the virus life cycle. Through this process, the p38 and ERK1-2 MAPK pathways are activated, which in turn promotes the nuclear export of viral ribonucleoprotein (vRNP), a crucial step in viral assembly and release. Remarkably, the NOX4 oxidase was the major contributor to the virus-induced oxidative stress and the enhancement of viral replication in murine primary airway epithelial cells and human lung cancer cell lines. Indeed, it was found that the expression of NOX4 was increased during cell infection, while inhibition of NOX4 activity blocked ROS increase, the phosphorylation of MAPK, the nuclear export of the vRNP and the viral release [138].

In addition, Garcia and colleagues [139] shed light on the interactions between COVID-19 and the DDR pathway and suggested DDR-associated kinase inhibitors as potent blockers of SARS-CoV-2 replication. Indeed, the authors screened a library of pharmacological compounds to find antiviral medicines that are specific to SARS-CoV-2 and found that virus cytopathic impact in human epithelial cells was inhibited by 34 of 430 protein kinase inhibitors that are in different phases of clinical trials. For example, berzosertib, a selective inhibitor of serine/threonine-protein kinase ATR, that is already in phase 2 clinical trials for solid tumors [140], prevented SARS-CoV-2 replication at the post-entry stage and had significant antiviral action in various cell types.

Importantly, a recent study has shown how COVID-19 damages the heart, opening the opportunity for new COVID-19 treatments [141]. In that study, the authors investigated the transcriptome landscape of cardiac tissues collected from SARS-CoV-2 infected patients and controls. Transcriptomics analysis showed upregulation of DNA damage and repair-related genes in the cardiac tissues of COVID-19 patients. In addition, the presence of DNA damage in the same tissues of SARS-CoV-2 patients was further confirmed using γH2AX immunostaining, an established methodology for detecting DNA damage.

### 3.3. COVID-19 and Oxidative Stress

Oxidative stress is defined as a dangerous state caused by the imbalance between the production and the accumulation of ROS [142]. ROS are highly reactive molecules that trigger rapid chain reactions and cause oxidative damage to macromolecules, such as lipids, proteins, carbohydrates, and nucleic acids, thus affecting various cellular functions. Previous data have shown that increased levels of oxidative stress participate in the onset and progression of many diseases, such as cancer and autoimmunity [143]. On the other hand, very low levels of oxidative stress result in the induction of reductive stress and the occurrence of pathologies ranging from cancer to cardiomyopathy [144].

A growing number of studies have shown that oxidative stress plays a key role in viral infections, such as SARS-CoV-2 [145,146,147]. Indeed, a primary characteristic of viral infection is an imbalance of redox equilibrium in the body [148]. By creating an excess of ROS and a shortage of reduced glutathione (GSH), the virus manipulates the host cell machinery to put the cell into an oxidative stress state, which creates favorable conditions for viral reproduction. Interestingly, the excess reactive oxygen/nitrogen species (RONS) synthesis and the abnormal cellular antioxidant-oxidant balance have been implicated in the pathogenesis of respiratory viral infections, such as SARS-CoV-2 [145].

Other studies suggested that oxidation of thiols to disulfides, because of oxidative stress, might boost SARS-CoV-2 spike protein affinity for the ACE2 receptor that is responsible for the degradation of the vasoconstrictor Angiotensin II (Ang II) to the vasodilator Angiotensin 1–7 (Ang 1–7) and so increase the severity of SARS-CoV-2 infection [149,150]. Because SARS-CoV-2 binding to the ACE2 receptor reduces the enzyme’s catalytic activity, i.e., the conversion of Ang II to Ang 1–7, the nicotinamide dinucleotide phosphate (reduced form, NADPH) oxidase activity may also rise in SARS-CoV-2 infected patients, resulting in an increase of oxidative stress [151]. Moreover, lower levels of the antioxidant glutathione enhance cellular oxidative stress, which is linked to a variety of diseases and immunological dysfunctions that increase viral infection susceptibility, such as uncontrolled SARS-CoV-2 infection [152]. We also have to mention that uncontrolled replication causes oxidative damage in the lungs, increasing viral load and consequently the severity of the virus infection [153]. Moreover, since the membrane antioxidant vitamin D has been reported to improve immunity and protect against respiratory illness, a recent study proposed that there is a connection between vitamin D levels and COVID-19 susceptibility [154,155].

Since oxidative stress might directly or indirectly affect the progression and outcome of SARS-CoV-2 infection, it is an emerging target in the battle against viral infection. Indeed, many studies have reported the use of antioxidants, such as N-acetylcysteine (NAC), GSH, polyphenols, and selenium, in the treatment of viral infections [156,157]. For example, a recent study has shown that NAC, due to its participation in the synthesis of glutathione, by boosting T cell response, preventing the depletion of the T cells and reducing inflammation, could be a promising medication to treat COVID-19 infection [158,159]. Other studies demonstrated that the in vitro or in vivo administration of GSH derivatives inhibited Sendai virus and Herpes Simplex Virus 1 (HSV-1) replication, without inducing toxic effects [160,161]. Moreover, previous data have shown that polyphenol components derived from pistachios kernels (the raw kernels of the pistachio nut) exhibited antiviral effects, with resveratrol, a stilbene derived from a variety of plants, being the best anti-HSV nutraceutical agent [162,163]. In addition, selenium-based nanoparticles have emerged as a promising approach in the treatment of influenza [164].

Importantly, in a recent study, the authors measured redox biomarkers and DNA damage levels for 14 days in hospitalized COVID-19 patients. Maximal levels of malondialdehyde, a biomarker of lipid peroxidation and oxidative stress, were observed at the time of hospitalization, rapidly dropping during the time-course analyzed, while 8-hydroxy-2′-deoxyguanosine (8-OHdG) levels, an important byproduct of oxidative DNA damage, peaked at 7 days after hospitalization [165]. Another study evaluated the presence of guanine oxidized species in COVID-19 hospitalized patients [166]. The authors reported that the levels of the DNA and RNA guanine oxidized species were higher in the serum of non-surviving COVID-19 patients than in those that survived, suggesting that oxidative DNA damage could be a predicting factor of death by COVID-19.

### 3.4. SARS-CoV-2 Vaccination, the Immune System and the DDR Network

The DDR network and the immune system are the main mechanisms that act together favoring the proper function of various organisms [110]. Indeed, several studies have shown that activated DDR induces immune responses, usually via a cGAS/STING-mediated pathway [167,168,169], while the activated immune system induces the DDR network through the generation of oxidative stress and the resulting DNA damage [170,171]. To further explore the interplay between these two systems, Ntouros and colleagues [172] investigated the effect of an acute immune challenge on the DDR system, using SARS-CoV-2 vaccination as an in vivo model of acute inflammation. They found that 24 h after SARS-CoV-2 vaccination (Comirnaty, Pfizer-BioNTech), peripheral blood mononuclear cells (PBMCs) of healthy individuals showed a transient increase of type I IFN, combined with elevated oxidative stress and accumulation of DNA damage; vaccination did not influence the DNA repair capacity of PBMCs. All these parameters resumed regular levels a few days later. Collectively, these data show that SARS-CoV-2 vaccination, as an acute immune stimulant, successfully triggers the DDR network. Moreover, the cytokine profile of the vaccinated individuals reveals a distinct interleukin 15, interferon gamma and IP10/CXCL10 signature, which correlates with effective immune activation [173].

A growing number of studies have shown that older people are characterized by decreased antibody response to SARS-CoV-2 vaccination [174]. To elucidate the link between aging and the response to SARS-CoV-2 vaccination, a recent report analyzed oxidative stress and accumulation of DNA damage in aged individuals before and after vaccination [175]. They found that after SARS-CoV-2 vaccination (Comirnaty, Pfizer-BioNTech, New York, NY 10017, USA), individual titers of anti-Spike-Receptor Binding Domain (S-RBD)-IgG antibodies and the neutralizing capacity of circulating anti-SARS-CoV-2 antibodies inversely correlated with the corresponding pre-vaccination oxidative stress status and the DNA damage levels observed in PBMCs. Together, these results suggest that humoral immune responses to SARS-CoV-2 vaccination may be weaker when immune cells are under oxidative and/or genotoxic stress, conditions that are common in the elderly.

## 4. Conclusions

Data present in this report demonstrate that infection with SARS-CoV-2 activates the DDR network in various ways (Figure 2). Indeed, in severe COVID-19 patients, the SARS-CoV-2-induced abnormal activation of the immune system triggers the induction of oxidative stress, which in turn causes damage to DNA, thus activating the DDR network. Moreover, SARS-CoV-2 can induce the generation of micronuclei containing DNA damage. Both the formation of micronuclei that initiate inflammatory gene expression, thus alerting the immune system to the presence of damaged cells, as well as the recognition of DNA damage in the micronuclei, which leads to the upregulation of the γH2AX and p53 components, result in the activation of the DDR network. Last but not least, following the SARS-CoV-2-induced inhibition of the TRF2 subunit of the Shelterin system, cells lose the protective activity of Shelterin, telomeres are no longer hidden from DNA damage surveillance, and chromosome ends are processed by DNA repair pathways, thus resulting in telomere shortening and the activation of the DDR network through the induction of the DNA damage sensing ATR kinase.

## Figures and Tables

**Figure 1 vaccines-10-01764-f001:**
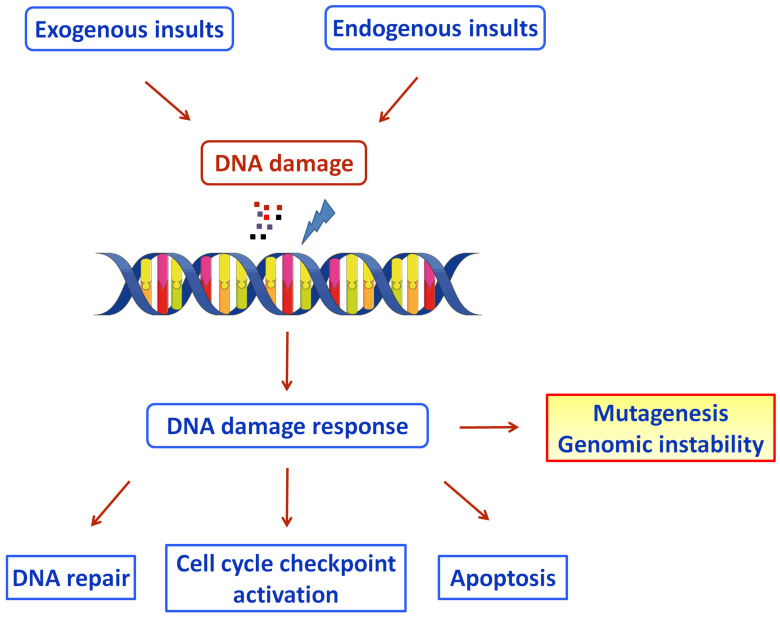
Schematic diagram of the DDR pathways activated by exogenous and endogenous insults (Figure was generated using images assembled from Servier Medical Art, https://smart.servier.com, accessed on 11 August 2022).

**Figure 2 vaccines-10-01764-f002:**
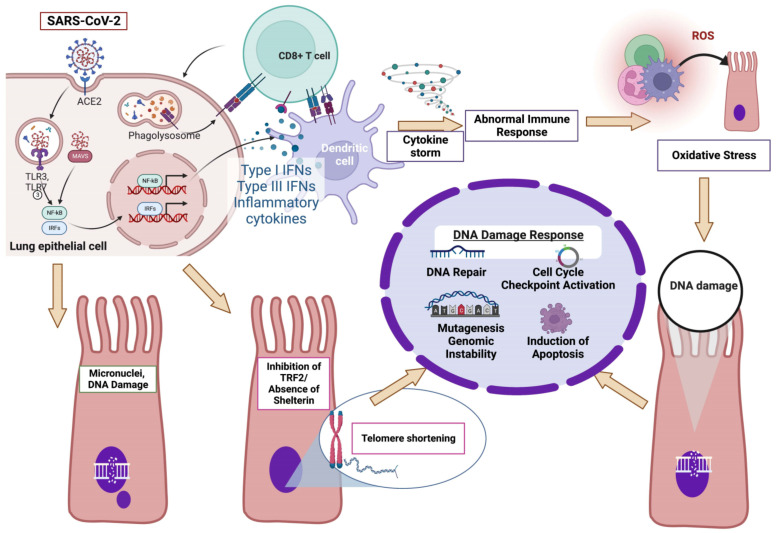
SARS-CoV-2-induced abnormal activation of the DDR network: a proposed model (for details see “Conclusions”; Created with BioRender.com, accessed on 5 September 2022).

## Data Availability

The data presented in this study are openly available in the reference section.

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
