# Peer review of "Delineating the SARS-CoV-2 Induced Interplay between the Host Immune System and the DNA Damage Response Network"

_vaccines, 2022, doi:10.3390/vaccines10101764_

Round 1

Reviewer 1 Report

The review is a fresh take on DNA mutations caused by the renowned COVID-19 infection. However, the concern of covid19 resulting in lower immune response is quite generic.

1. The titles in the Figure 2 are not clear, and describe the mechanism involved in breif as a figure legend.

2. Statements in Line 114-116 ( about secretion of INF-gamma) are confusing.

3. Many typ errors needs to be corrected. (For example: line 236 (typos), heart and cardiovascular disease? in line 214).

4. Any ongoing clinical studies to study DNA damage in covid-19 patents? If yes. please include them.

Reviewer 2 Report

This paper is an interesting oveview of the impact of DNA damage response network in SARS-CoV-2 infection/vaccination which is a field in constant development.

Author Response

We thank the Reviewer #2 for his/her positive comments on our manuscript

Reviewer 3 Report

I read the article “Delineating the SARS-CoV-2 induced interplay between the host immune system and the DNA damage response network” that focuses on dysfunction of immune system and SARS-CoV-2. This review is well-structured and the topic is dealt with in an exhaustive way. The title is appropriate. Therefore, I think that this article is suitable for publication in its current version.

Author Response

We thank the Reviewer #3 for his/her positive comments on our manuscript

Reviewer 4 Report

The paper reviews recent works on the interaction of SARS-CoV-2 immune response with hyper-inflammation, DNA damage and repair, and oxidative stress. These topics are in the main stream of the current research. The paper is representative for the current literature and basically well written. However, in many cases it is reduced to the references to some observations without the analysis of underlying mechanisms, contrary to what is stated in the abstract (line 29). In particular:

1.       Section 1.1: “Evidence supports that aberrant immune response to SARS-CoV-2 has been mainly attributed to impaired interferon (IFN) signaling, hyper-inflammation and delayed adaptive immune responses [6].” It would be useful to discuss in more detail, with the corresponding molecular mechanisms, how virus down-regulates interferon. Here and in what follows some schematic representations of these molecular mechanisms would help the readers.

2.       Section 1.2. As before, it is stated that impaired interferon production is observed in severe cases, but what are the mechanisms of this impaired production? Even though such mechanisms are not completely elucidated, some data and plausible hypothesis might be presented.

3.       Section 1.3. Similar for cytokine storm, “Normally, the above would favor virus clearance. However, in the case of moderate or severe COVID-19, high viral load and/or individual immunogenetic factors alter the immune landscape; …” What are the mechanisms by which virus shifts the equilibrium between inflammatory and anti-inflammatory cytokines, what is the role of T reg cells and macrophages, are there aberrant immune cells, etc?

4.       Section 2. There is only a short indication of the interaction of infection with cancer: “As expected, their susceptibility to longer and severer COVID-19 course is mainly attributed to their systemic immunosuppressive status, derived either by cancer itself or the anticancer treatment [55].” This is not sufficient for a review on molecular mechanisms.

5.       Figure 1, caption “Molecular pathways …” I am not sure that this scheme can be called molecular pathways, these are not molecules but processes.

6.       Line 315. How DNA damage is related to virus clearance?

7.       Line 368. How oxidative stress creates favorable conditions for virus replication?

In conclusion, the topic of the paper is timely and presented material is useful. More detailed discussion of underlying molecular mechanisms would reinforce the paper.

Round 2

Reviewer 4 Report

The authors addressed my comments. The paper can be accepted.